# CircKIF4A Is a Prognostic Factor and Modulator of Natural Killer/T-Cell Lymphoma Progression

**DOI:** 10.3390/cancers14194950

**Published:** 2022-10-09

**Authors:** Rongfang He, Wei Wen, Bibo Fu, Renjie Zhu, Guanjun Chen, Shenrui Bai, Xi Cao, Hua Wang

**Affiliations:** 1The First Affiliated Hospital, Department of Pathology, Hengyang Medical School, University of South China, Hengyang 421001, China; 2Department of Hematological Oncology, Sun Yat-sen University Cancer Center, State Key Laboratory of Oncology in South China, Collaborative Innovation Center for Cancer Medicine, Guangzhou 510060, China; 3East Hospital Affiliated to Tongji University, Shanghai 200120, China

**Keywords:** circKIF4A, NKTL, miR-1231, PDK1, BCL11A, poor prognosis

## Abstract

**Simple Summary:**

CircKIF4A is significantly upregulated in NKTL cell lines and its upregulation correlates with the poor prognosis of NKTL. CircKIF4A regulates PDK1 and BCL11A expressions by sponging miR-1231. Our data indicated that circKIF4A is oncogenic in NKTL and that it is a predictor of poor prognosis of NKTL.

**Abstract:**

**Background:** Natural killer/T-cell lymphoma (NKTL) is difficult to treat. Circular RNAs (circ RNAs) have been implicated in tumorigenesis. However, the function of circKIF4A in NKTL has not been investigated. **Methods:** QPCR analysis was used to compare circKIF4A levels in NKTL cell lines versus normal cell lines. Kaplan–Meier survival analysis was used to assess the effect of circKIF4A on the prognosis of NKTL. The correlation between clinicopathological features and circKIF4A expression was examined using cox regression analysis. Luciferase reporter, RNA immunoprecipitation and immunohistochemistry assays were also used to investigate the mechanisms of circKIF4A in NKTL. **Results:** Our analyses revealed that circKIF4A is significantly upregulated in NKTL cell lines and that its upregulation correlates with the poor prognosis of NKTL. The silencing of circKIF4A significantly suppressed glucose uptake and lactate production in NKTL cells. Moreover, we showed that circKIF4A, PDK1, and BCL11A bind miR-1231 and that circKIF4A regulates PDK1 and BCL11A expressions by sponging miR-1231. **Conclusions:** During NKTL progression, circKIF4A regulated PDK1 and BCL11A levels by sponging miR-1231. Our data indicated that circKIF4A is oncogenic in NKTL and that it is a predictor of poor prognosis of NKTL.

## 1. Introduction

Natural killer/T-cell lymphoma (NKTL), a lymphoproliferative disease associated with Epstein–Barr virus (EBV), is a highly aggressive subtype of non-Hodgkin’s lymphoma [1]. NKTL stains positive for CD2, CD56, cytoplasmic CD3ε, granzyme B, perforin, TIA1, and EBV-encoded RNA (EBER). The expression of EBER is the most important indicator of NKTL. The initial treatment for early-stage (stage I/II) NKTL is chemotherapy and radiation therapy, which can be given simultaneously or sequentially. In stage III/IV disease, anthracycline-based chemotherapy is no longer indicated, and regimens containing a combination of chemotherapeutic agents, such as L-asparaginase and radiation therapy, are the current treatment of choice. Autologous hematopoietic stem cell transplantation (HSCT) has also been used to treat this disease. Programmed cell death protein 1, anti-CD30 and CD38 monoclonal antibodies, and immunotherapies against EBV-related targets are increasingly used in the treatment of NKTL [2,3]. Although several biomarkers have been identified and applied in the treatment of NKTL, it is characterized by high recurrence and poor prognosis [4]. Hence, effective therapeutic targets for NKTL treatment are urgently needed.

CircRNAs, which are highly conserved non-coding RNAs, are predominantly localized in the cytoplasm, possess a unique covalently closed loop structure and are defined by high abundance and stability [5]. CircRNAs have a variety of functions, including sponging miRNAs, acting as competing endogenous RNAs (ceRNAs), binding to proteins, regulating protein translation, and regulating selective splicing [6]. Several studies have implicated the function of circRNAs in the occurrence and development of cancer. Cen et al. reported that hsa_circ_0005358 isolates PTBP1, preventing it from stabilizing CDCP1 mRNA, thereby suppressing CDCP1 protein translation and inhibiting cervical cancer metastasis [7]. It was reported that circPPP6R3 enhanced renal clear cell carcinoma progression by sponging miR-1238-3p, which leads to CD44 upregulation [8]. CircROBO1 mediates breast cancer carcinogenesis and liver metastasis via the circROBO1/KLF5/FUS feedback loop [9]. KIF4A is essential for alignment of chromosome, which participates in the assembly of centrosome, the formation of spindle, and DNA damage repair in mitosis anaphase [10,11,12]. By functioning as ceRNA, circKIF4A promotes ovarian cancer progression by sponging miR-127 [13]. CircKIF4A has also been proposed as a therapeutic target and prognostic biomarker in triple-negative breast cancer [14]. Nevertheless, the significance of circKIF4A in NKTL has yet to be established. Our findings suggested that circKIF4A is significantly upregulated in NKTL cells and that high circKIF4A levels are predictors for poor progression-free survival (PFS) and overall survival (OS) in NKTL patients. The analysis of the role of circKIF4A in NKTL revealed that by functioning as a ceRNA for miR-1231, it modulates the expression of PDK1 and BCL11A in NKTL. Thus, circKIF4A has prognostic and therapeutic potential against NKTL.

## 2. Methods

### 2.1. Cell Cultures and Transfection

Mycoplasma-free cell lines (the NK cell lines NK01 and NK02 and the NKTL cell lines SNK-1, HANK-1, NK-YS, KHYG-1, NKTL-7, NKTL-16, NKTL-22, and NKTL-29) were procured from the American Type Culture Collection. Cell identities were authenticated using DNA fingerprinting. Cells were transfected using Lipofectamine 2000 (Invitrogen, Waltham, MA, USA). SiRNAs targeting circKIF4A were synthesized by GenePharma (Shanghai, China). MiR-1231 mimics as well as inhibitors were obtained from GeneCopoeia (Rockville, MD, USA).

### 2.2. Real-Time Quantitative PCR Analysis (qPCR)

The isolation of total RNA was carried out using TRIzol (Invitrogen, Waltham, MA, USA). The isolation of cytoplasmic and nuclear fractions was carried out by NE-PER™ cytoplasmic and nuclear extraction reagents (Thermo Scientific, Waltham, MA, USA). QPCR was performed using SYBR premix Ex Taq™ (Takara, Shiga, Japan) and an all-in-One™ miRNA qPCR detection kit (GeneCopoeia, Rockville, MD, USA) on a Bio-Rad IQTM5 multicolor real time PCR system (Hercules, CA, USA). The primer of miR-1231 was purchased from Ruibiotech (Guangzhou, China). Thermal cycling was as follows: 95 °C, 5 min; 40 cycles of amplification (94 °C, 20 s; 58 °C, 25 s; 72 °C, 30 s). The sequences of qPCR primers: circKIF4A, 5′-GAGGTACCCTGCCTGGATCT-3′ (forward) and 5′-TGGAATCTCTGTAGGGCACA-3′ (reverse); KIF4A, 5′-TACTGCGGTGGAGCAAGAAG-3′ (forward) and 5′-CATCTGCGCTTGACGGAGAG-3′ (reverse); PDK1, 5′-CATGTCACGCTGGGTAATGAGG-3′ (forward) and 5′-CTCAACACGAGGTCTTGGTGCA-3′ (reverse); BCL11A, 5′-ACAAACGGAAACAATGCAATGG-3′ (forward) and 5′-TTTCATCTCGATTGGTGAAGGG-3′ (reverse); GAPDH, 5′-GGAGCGAGATCCCTCCAAAAT-3′ (forward) and 5′-GGCTGTTGTCATACTTCTCATGG-3′ (reverse). Sequences for siRNAs: si-circNC, UUCUCCGAACGUGUCACGUTT; si-circKIF4A#1, GAUCUAUAACGUAUUAAUATT; si-circKIF4A#2, GCCUGGAUCUAUAACGUAUTT.

### 2.3. RNase R Resistant Assay

A total of 5 µg of RNA were treated with RNase R (2 U/μg; Beyotime, R7092M, Nantong, China) or mock for 30 min at 37 °C. Remaining RNAs, including circRNA, were quantified using qPCR.

### 2.4. Actinomycin D Digestion

An number of 1 × 10^5^ cells were inoculated into 6-well plates and treated with actinomycin D (2 mg/L; Cat No, Sigma, St. Louis, MO, USA) immediately. Then, the treated cells at the indicated times (8 h, 16 h, and 24 h) were obtained for the quantification of their circular (circKIF4A) and linear forms of KIF4A mRNA by qPCR.

### 2.5. Luciferase Reporter Assay

The circKIF4A sequences, including the miR-1231 binding sites (CGUCGACAGGCGGGUCUGUG), were inserted into the pGL3 luciferase vector (Promega, USA) immediately downstream of luciferase. Mutations in the miR-1231 seed-region were conducted with Fast Site-Directed Mutagenesis Kit (Tiangen, Beijing, China). The 3′-UTR of BCL11A and PDK1 including the miR-1231 binding sites (CGUCGACAGGCGGGUCUGUG;CGUCGACAGGCGGGUCUGUG) was inserted into the pGL3 luciferase vector. Mutations in the miR-1231 seed-region served as a mutant control.

Cells (5 × 10^3^) were planted in plates and co-transfected with respective vectors along with miR-1231 mimics or inhibitors. After incubation for 48 h, luciferase activities were measured using a dual-luciferase reporter assay system (Promega, Madison, WI, USA).

### 2.6. RNA Immunoprecipitation (RIP)

A total of 3 × 10^6^ cells were seeded on a 6cm plate, and co-transfection with MS2bp-GFP, along with MS2bs-circKIF4Amt (the miR-1231 binding sites were mutated), MS2bs-circKIF4A, or MS2bs-Rluc, were carried out for 48 h. Then, four plates of cells in each group were harvested to perform RIP using a Magna RIP RNA-binding protein immunoprecipitation kit (Millipore, Burlington, MA, USA) following the manufacturer’s protocols. RNA complexes were purified, and miR-1231 levels were determined.

Ago2 RIP was carried out using an anti-Ago2 antibody (Millipore, Burlington, MA, USA) following the manufacturer’s protocols. Then, RNA was purified after which circKIF4A, PDK1, and BCL11A levels were assessed.

### 2.7. Glucose Uptake and Lactate Production

Cell were transfected with negative control siRNA or circKiF4A siRNA and then were harvested for the detection of glucose uptake (J1341, Promega, Madison, WI, USA) and lactate production (J5021, Promega, Madison, WI, USA) according to the manufacturer’s instructions.

### 2.8. Immunohistochemistry (IHC)

IHC was performed using primary antibodies against PDK1(ab202468, 1:100, Abcam, Cambridge, UK), BCL11A (ab191401, 1:100, Abcam, Cambridge, UK), Ki67(ab16667, 1:100, Abcam, UK), and HRP-conjugated secondary antibodies (Thermo Scientific, Waltham, MA, USA). Antigen retrieval was carried out for 5 min at 120 °C in a pressure cooker and followed by antibody incubation overnight at 4 °C. Suitable positive tissue controls were used. BCL11A and PDK1 levels were determined as a % of total tumor cell populations per 1 mm core diameter. The proportions of BCL11A- and PDK1-positive cells were determined in three representative high-power fields of individual samples. The examination of individual samples was blindly conducted by at least two pathologists, with scores of <2 and ≥2 indicating low and high expression, respectively.

### 2.9. Patients Samples and Ethical Standards

Between January 2008 and September 2015, 227 NKTL patients were diagnosed at the Sun Yat-sen University Cancer Center. In the same period, 68 NKTL patients were recruited at the first affiliated hospital of Hengyang Medical School, University of South China. Patient diagnosis was based on the WHO’s guidelines on the classification of lymphoid and hematopoietic tumors (4th edition, 2016). Diagnostic confirmation was attained through positive results of EBER in situ hybridization. Induction chemotherapy was the primary treatment option, followed by consolidative radiotherapy. The determination of complete remission was based on non-Hodgkin’s lymphoma response criteria (Cheson et al., 1999). This study was approved by the Ethics Committee of the SYSUCC and the first affiliated hospital of Hengyang Medical School, University of South China, and performed in accordance with the Declaration of Helsinki. Written informed consent was obtained from all patients before participation in this study.

### 2.10. Statistical Analysis

SPSS version 19.0 was used for analyses. Between-group comparisons were carried out using t and Pearson *χ^2^* tests. Survival analyses were carried out using Kaplan–Meier plots and log-rank tests. Unless otherwise stated, data are shown as mean ± SD for *n* = 3. A two-sided *p*-value of <0.05 was considered statistically significant. Each experiment was repeated at least three times.

## 3. Results

### 3.1. CircKIF4A Was Upregulated in NKTL and Correlated with Poor Survival

The QPCR analysis of circKIF4A expression revealed that compared with the normal NK cell lines (NK01 and NK02), circKIF4A was upregulated in the eight NKTL cell lines (SNK-1, HANK-1, NK-YS, KHYG-1, NKTL-7, NKTL-16, NKTL-22, and NKTL-29, Figure 1A). RNase R resistance analysis and digestion analysis using actinomycin D confirmed that circKIF4A was more stable than linear KIF4A mRNA (Figure 1B,C).

Kaplan–Meier survival analysis indicated that NKTL patients with high circKIF4A levels had poorer OS (*p* < 0.001, Figure 2A) and progression-free survival (PFS) (*p* < 0.0001, Figure 2B). Moreover, NKTL patients with higher Eastern Cooperative Oncology Group (ECOG) scores (*p* < 0.0001, Figure 2C,D), increased high lactate-dehydrogenase (LDH) levels (*p* < 0.0001, Figure 2E,F), regional lymph node invasion (*p* < 0.005, Figure 2G,H), localized invasion (*p* < 0.0001, Figure 2K,L), advanced stage (*p* < 0.0001, Figure 2M,N), and later clinical stage (*p* < 0.0001, Figure 2O,P), had significantly poorer OS and PFS. However, the presence or absence of B symptoms (non-specific systemic symptoms usually including unintentional weight loss, fever, and night sweats, which could be associated with the potential presence of lymphoma) was not associated with OS and PFS (*p* > 0.05, Figure 2I,J). Additionally, univariate and multivariate COX regression analyses showed that circKIF4A expression was an independent prognostic indicator of NKTL (Table 1).

### 3.2. CircKIF4A Silencing Inhibits Glucose Uptake and Lactate Production of NKTL Cells

QPCR analysis revealed that circKIF4A levels were reduced in HANK-1 and KHYG-1 cell lines after transfection with si-circKIF4A (Figure 3A,B). Moreover, we observed that circKIF4A silencing significantly suppressed glucose uptake and lactate production in NKTL cells (Figure 3C,D). These results show that silencing circKIF4A inhibits glycolysis of NKTL cells, which is a major source of energy metabolism in tumor cell growth.

### 3.3. CircKIF4A Functions as a Sponge for miR-1231 in NKTL

The QPCR analysis of the subcellular localization of circKIF4A revealed that it mainly localizes in the cytoplasm. Because miRNA also predominantly localizes in the cytoplasm, this suggested that circKIF4A may be a miRNA sponge (Figure 4A,B). QPCR analysis revealed the reduced miR-1231 levels in NKTL cell lines (Figure 4C). Next, using Targetscan, we predicted the binding sites of miR-1231 in the circKIF4A sequence (Figure 4D). Luciferase reporter assays revealed that co-transfecting wild-type circKIF4A luciferase reporter with miR-1231 mimics suppressed luciferase intensity and that the mutant luciferase reporter did not have this effect (Figure 4E,F). Additionally, we found that miR-1231 was primarily enriched in the MS2bs-circKIF4A group rather than in the control group and MS2bs-circKIF4A-mut group (Figure 4G), implying that circKIF4A directly interacts with miR-1231 and that it may function as a miR-1231 sponge.

### 3.4. CircKIF4A Regulates NKTL through BCL11A and PDK1

To investigate the mechanism of circKIF4A in NKTL, we used TargetScan to predict miR-1231 binding sites and identified potential miR-1231 binding sites in the 3′-UTRs of BCL11A and PDK1 (Figure 5A,B). This identified BCL11A and PDK1 as potential miR-1231 targets. Co-transfecting miR-1231 mimics with wild type PDK1 and BCL11A luciferase reporters revealed reduced luciferase intensity when compared with mutant PDK1 and BCL11A luciferase reporters (Figure, 5C,D). Moreover, transfecting miR-1231 into HANK-1 and KHYG-1 cells suppressed the expression of BCL11A and PDK1 (Figure 5E). Together, these data suggested BCL11A and PDK1 as miR-1231 targets. Next, the RIP analysis of the AGO2 protein complex revealed that circKIF4A knockdown abrogated circKIF4A enrichment by Ago2 and increased the aggregation of BCL11A and PDK1 (Figure 5F), implying that circKIF4A functions as a ceRNA and competes with PDK1 and BCL11A. Transfection with miR-1231 reduced glucose uptake and lactate production by NKTL cells and this effect was countered by PDK1 and BCL11A (Figure 6A,B). The analysis of the expression of PDK1 and BCL11A in NKTL patients revealed that they were highly expressed and positively correlated (Figure 6C). Quantified results of the ERBR, Ki-67, PDK1, and BCL11A were shown in Appendix A. Together, these findings imply that circKIF4A influences the progression of NKTL via the circKIF4A-miR-1231-PDK1/BCL11A axis.

## 4. Discussion

NKTL is a lymphoproliferative disorder associated with the Epstein–Barr virus (EBV). The disease is most frequently reported in Asia and Latin America [15]. NKTL can present as nasal congestion, epistaxis, purulent and/or bloody rhinorrhoea, chronic sinusitis, and necrotic lesions of the nose or hard palate [16]. Although several strategies have been developed for NKTL therapy, a large proportion of patients fail to respond to treatment or experience recurrence [17].

Previous studies indicate that circRNAs influence tumorigenesis by regulating the cell cycle, signal transduction, and transcription [18]. We found that circKIF4A levels were upregulated in NKTL and significantly correlates with poor prognosis. By silencing circKIF4A, we observed that glucose uptake and lactate production were inhibited in NKTL cells and the expressions of PDK1 and BCL11A were influenced. Our findings indicate that circKIF4A is oncogenic in NKTL and that it is a potential prognostic marker for NKTL. Several studies have reported that circRNAs can exert their pro- or anti-cancer effects by sponging miRNAs [19,20]. In glioma, circKIF4A is reported to promote Wnt5a expression by sponging miR-139-3p, thus promoting cell migration and invasion and disease progression [21]. The circKIF4A-miR-1231-GPX4 axis was reported to promote papillary thyroid cancer progression [22]. However, the role of circKIF4A in NKTL is unclear. Here, we identified a binding site for miR-1231 in circKIF4A, indicating that circKIF4A may promote NKTL progression by sponging miR-1231.

MiR-1231 has been implicated in tumorigenesis. It was reported that miR-1231 downregulation suppresses prostate cancer cell proliferation, migration, and invasion by targeting EGFR [23]. Low miR-1231 expression has been proposed as a marker of clinical stage, lymph node invasion, and poor prognosis in ovarian cancer [24]. PDK1 (Pyruvate dehydrogenase kinase 1) is a serine-threonine kinase and belongs to the AGC kinase family. PDK1 is a key component of phosphatidylinositol 3-kinase (PI3K) pathway signaling, activating multiple downstream effectors and thereby coordinating intra-cellular and extra-cellular signaling. As a glycolytic protein it influences cancer cell proliferation, metastasis [25,26], and cell motility [27]. PDK1 also protects cells from apoptosis in response to hypoxia and oxidative stress [28,29,30]. BCL11A (B-cell lymphoma/leukemia 11A) gene is a cancer gene that encodes the C2H2 zinc finger transcription factor and a component of the BRAF complex, which has been shown to be associated with the regulation of cell proliferation, apoptosis, and transformation. It is mainly expressed in the brain, breast, immune cells, hematopoietic cells, hematopoietic stem cells, common lymphoid progenitor cells, B cells, and early T cell precursor cells [31,32]. Studies have demonstrated that BCL11A expression was increased in NKTL patients and NKTL cell lines and correlated with the poor prognosis of NKTL [33]. Meanwhile, our experiments showed that the expression of BCL11A was correlated with circKIF4A.

In summary, our data show that circKIF4A is highly expressed in NKTL and that it is a predictor of poor NKTL prognosis. We found PDK1 and BCL11A as targets of miR-1231 in NKTL, and the transfection of miR-1231 inhibits the expression of PDK1 and BCL11A. CircKIF4A potentially regulates the expression of PDK1 and BCL11A by sponging miR-1231. All in all, our findings indicate that circKIF4A may influence NKTL tumor growth through modulating PDK1 and BCL11A expression by functioning as a ceRNA for miR-1231. Therefore, circKIF4A may become a potential prognosis predictor and therapeutic targets for NKTL.

## Figures and Tables

**Figure 1 cancers-14-04950-f001:**
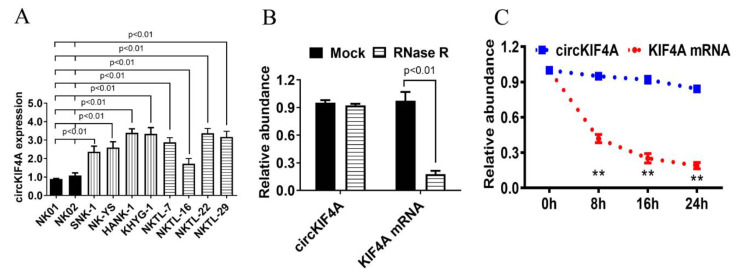
CircKIF4A is upregulated in NKTL. (**A**) CircKIF4A expression in the NKTL cell lines, SNK-1, NK-YS, HANK-1, KHYG-1, NKTL-7, NKTL-16, NKTL-22, and NKTL-29, and the normal NK cell lines, NK01 and NK02. (**B**) RNase R analysis of the circKIF4A and KIF4A mRNA. (**C**) Actinomycin D treatment was used to determine the stability of circKIF4A. ** indicates *p* < 0.01.

**Figure 2 cancers-14-04950-f002:**
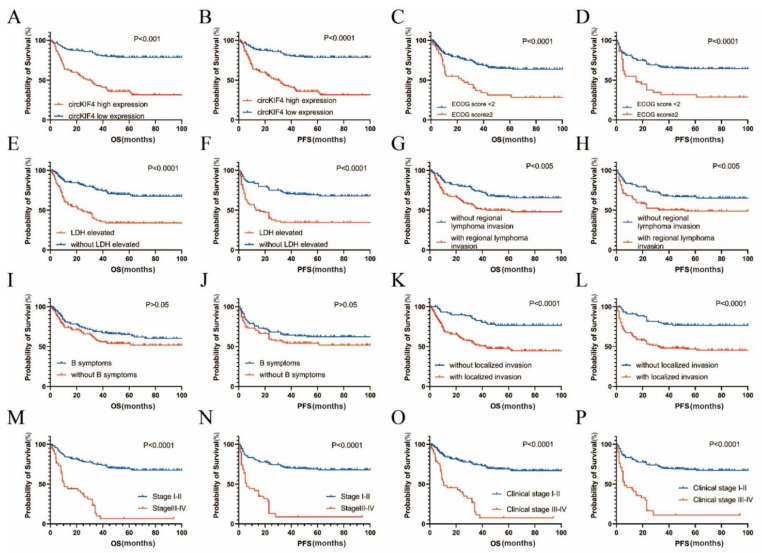
Correlation between clinicopathological features and circKIF4A expression and related survival rates in the 295 NKTL patients. (**A**,**B**) High circKIF4A expression levels correlated with poor OS and PFS. (**C**,**D**) ECOG scores of ≥2 correlated with poor OS and PFS. (**E**,**F**) Elevated LDH levels correlated with poor OS and PFS. (**G**,**H**) Regional lymph node invasion correlated with poor OS and PFS. (**I**,**J**) Presence or absence of B symptoms did not correlate with OS and PFS. (**K**,**L**) Localized invasion correlated with poor OS and PFS. (**M**,**N**) Patients with late stage exhibited poor OS and PFS. (**O**,**P**) Patients with a late clinical stage had poor OS and PFS.

**Figure 3 cancers-14-04950-f003:**
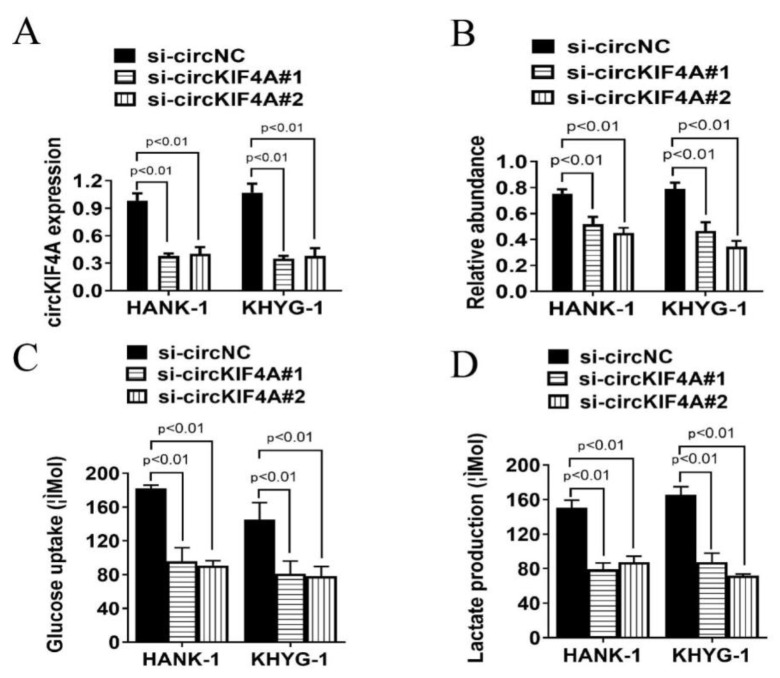
CircKIF4A silencing affected glucose uptake and lactate production in NKTL. (**A**,**B**) qPCR analysis of siRNA-mediated circKIF4A silencing. (**C**) CircKIF4A silencing associated with decreased glucose uptake levels. (**D**) CircKIF4A silencing correlated with reduced lactate levels. si-circNC, negative control siRNA; si-circKIF4A#1 and si-circKIF4A#2, two siRNAs designed to inhibit circKIF4A expression.

**Figure 4 cancers-14-04950-f004:**
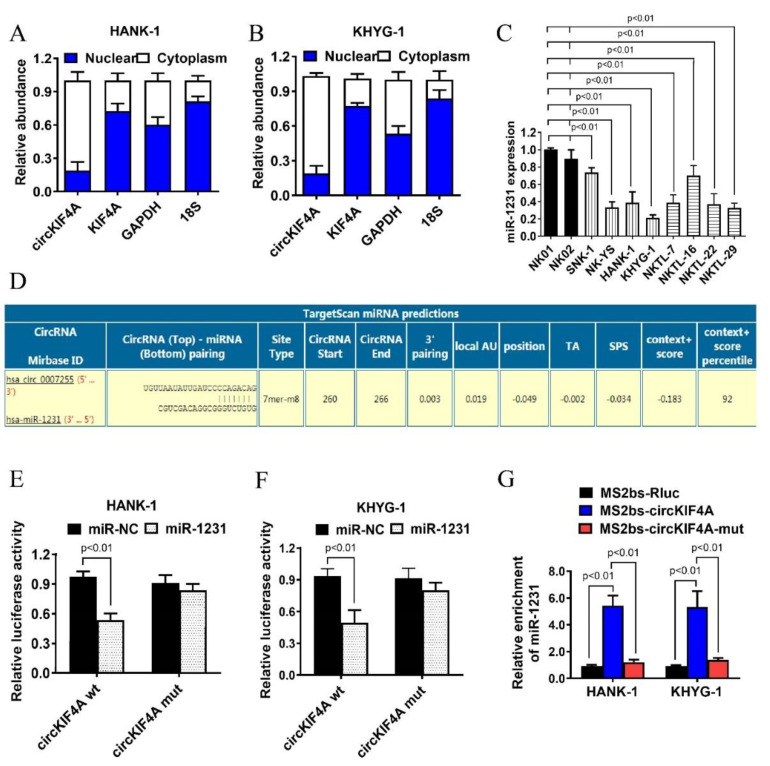
CircKIF4A functions as an miR-1231 sponge in NKTL. (**A**,**B**) Levels of the nuclear control (18S), cytoplasmic control (GAPDH), circKIF4A, and KIF4A. (**C**) MiR-1231 expression in the NKTL cell lines, SNK-1, NK-YS, HANK-1, KHYG-1, NKTL-7, NKTL-16, NKTL-22, and NKTL-29, and in the normal NK cell lines, NK01 and NK02. (**D**) Predicted miR-1231 binding site in circKIF4A. (**E**,**F**) Luciferase assay in HANK-1 and KHYG-1 cells transfected with miR-1231 mimics along with wild type or mutant circKIF4A luciferase reporters. (**G**) MS2-based RIP assay in cells transfected with MS2bs-circKIF4A, MS2bs-circKIF4A-mut, or control.

**Figure 5 cancers-14-04950-f005:**
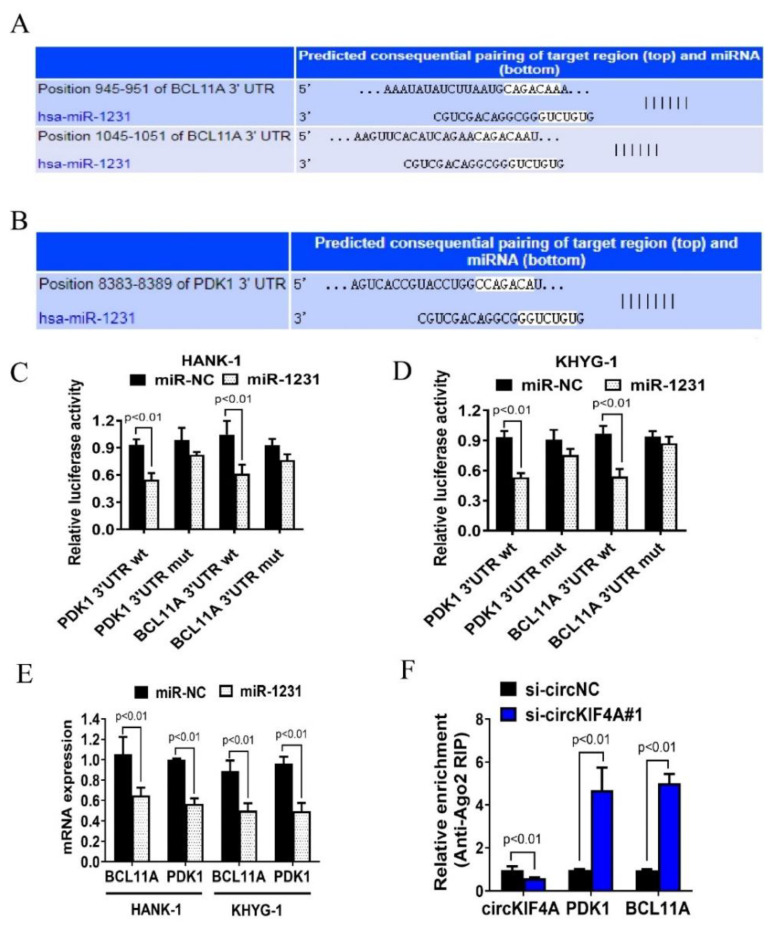
CircKIF4A influences NKTL via the circKIF4A-miR-1231-PDK1/BCL11A axis. (**A**,**B**) Predicted miR-1231 binding sites in the 3’ UTRs of BCL11A and PDK1. (**C**,**D**) Cells were transfected and subjected to the luciferase assay. (**E**) The expression of BCL11A and PDK1 was suppressed after transfecting HANK-1 and KHYG-1 cells with miR-1231. (**F**) Cells were transfected and subjected to Ago2-based RIP.

**Figure 6 cancers-14-04950-f006:**
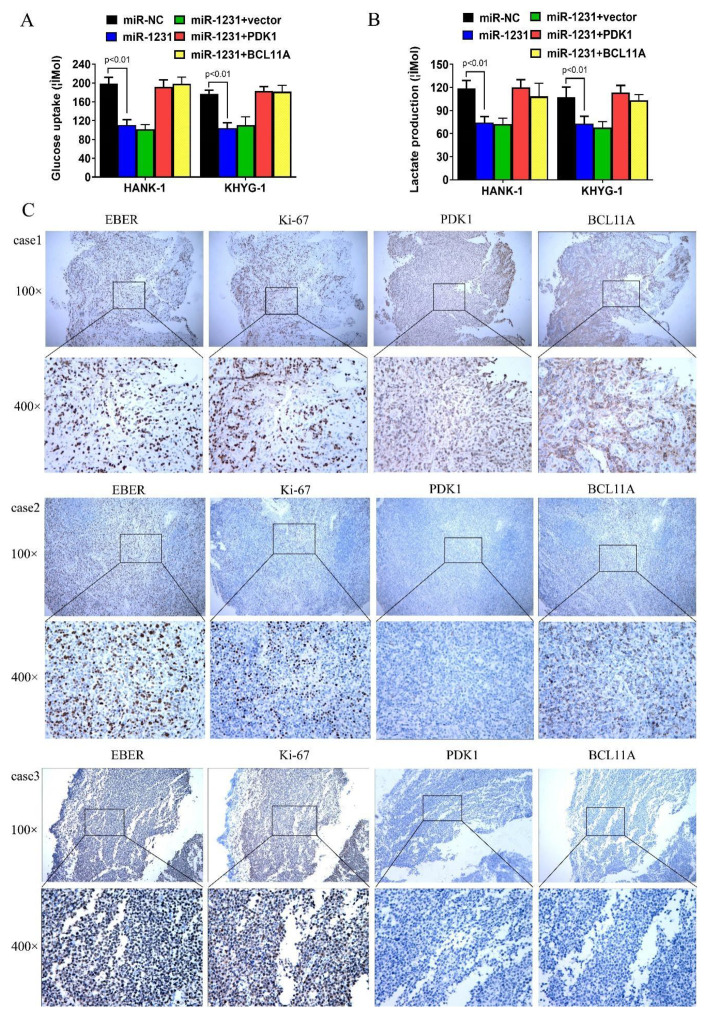
The relationship between miR-1231, BCL11A, PDK1, glucose uptake, and lactate production, and the association between PDK1 and BCL11A protein levels in NKTL. (**A**,**B**) BCL11A and PDK1 reversed the negative effects of miR-1231 on glucose uptake and lactate production. (**C**) PDK1 and BCL11A protein levels in NKTL. Case 1 was strongly positive for BCL11A and PDK1. Case 2 was weakly positive for BCL11A and PDK1. Case 3 was negative for BCL11A and PDK1. All cases were strongly positive for EBER and Ki-67.

**Table 1 cancers-14-04950-t001:** Univariate and multivariate COX analysis of OS in NKTL patients.

Variables	Hazard Ratio	Std. Err.	*p*	[95% Conf. Interval]	
Gender (male vs. female)	0.752	0.231	0.219	0.478	1.184
Age (<60 vs. ≥60)	1.333	0.262	0.274	0.797	2.229
clinical stage (I–II vs. III–IV)	5.575	0.223	<0.001 *	3.599	8.637
ECOG score (<2 vs. ≥2)	3.960	0.211	<0.001 *	2.618	5.991
circKIF4 (high expression vs. low expression)	0.606	0.208	0.016 *	0.403	0.912
LDH (<60U/L vs ≥60U/L)	2.962	0.210	<0.001 *	1.964	4.466
B symptoms (positive vs. negative)	1.379	0.210	0.122	0.918	2.073
regional lymph nodes invasion (negative vs. positive)	1.962	0.245	0.001 *	1.299	2.964
Multivariate analysis					
clinical stage (I–II vs. III–IV)	1.747	0.290	0.055	0.989	3.084
ECOG score (<2 vs. ≥2)	1.166	0.296	0.603	0.654	2.081
circKIF4 (high expression vs. low expression)	0.500	0.223	0.002 *	0.323	0.774
LDH (<60U/L vs ≥60U/L)	0.741	0.232	0.196	0.471	1.167
regional lymph nodes invasion (negative vs. positive)	0.000	52.764	0.803	0.001	1.577

Abbreviations: OS: overall survival; ECOG: eastern cooperative oncology group; LDH: lactate dehydrogenase.* indicates *p* < 0.05.

## Data Availability

The datasets used and analyzed in the current study are available from the corresponding author on reasonable request.

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
