# Peer review of "CircKIF4A Is a Prognostic Factor and Modulator of Natural Killer/T-Cell Lymphoma Progression"

_cancers, 2022, doi:10.3390/cancers14194950_

Round 1

Reviewer 1 Report

The topic of the publication is very interesting and could bring new knowledge regarding NKTCL. However, the submitted manuscript has a number of shortcomings and lacks important information. In particular, the following should be mentioned:

1) The full name (circular RNA) should be used before the abbreviation circRNA. The same should be done in case of all other abbreviations used in the manuscript.

2) The physiological role of KIF4A should be mentioned.

3) Methods and material were not properly described:

a) PCR conditions and primers are missing.

b) The RNAse R supplier is not listed.

c) In the case of actinomycin D, cells were seeded in 6-well plates. However, there is no information whether the treatment was done immediately or whether the cells were first cultured at some time interval and then treated with actinomycin D.

d) How many cells were seeded for RNA immunoprecipitation and how long were they cultured before transfection?

e) RNA complex purification method and primers for miR-1231, PDK1 and BCL11A are missing.

f) MS2bs-circKIFAmt should be described/explained.

g) Antibodies used in the immunohistochemistry should be specified.

h) siRNA sequences are missing.

4) More detailed information about the patients and their therapy is missing. It is also unclear whether the samples were obtained before, during, or after therapy. If after, information about the treatment regimen should be given.

5) More information should be provided on RNAse R resistance analysis. How was the circularity of KIF4A confirmed?

6) In the case of Fig. 1C, the units are missing.

7) NKTL is not B-cell lymphoma. Why were B-symptoms used in the COX analysis?

8) The authors reported that COX multivariate analysis showed that circKIF4A expression, clinical stage, ECOG score, LDH levels and metastatic status were independent prognostic indicators of NKTL. However, p<0.05 was detected in case of multivariate analysis only for circKIF4. In addition, there are several values of 0.000 in Table 1.

9) Fig. 2: Units for OS, PFS (x-axis) and probability of survival (y-axis) are missing.

10) The reason why the authors studied glucose and lactate production should be explained and the methods of glucose and lactate production study should be described in the methodology section.

11) Fig. 3: It is not explained what NC, A#1 and A#2 mean.

12) The reason why the authors focused on miR-1231 should be explained.

13) MS2bs-circKIF4A-mut and PDK1 and BLC11A luciferase reporter mutations should be specified.

14) Fig.  6. More information about cases 1-3 should be provided. Samples were taken from 227 NKTCL patients. It is not clear what cases 1-3 mean.

Author Response

Review1

Comments and Suggestions for Authors

The topic of the publication is very interesting and could bring new knowledge regarding NKTCL. However, the submitted manuscript has a number of shortcomings and lacks important information. In particular, the following should be mentioned:

  • The full name (circular RNA) should be used before the abbreviation circRNA. The same should be done in case of all other abbreviations used in the manuscript.

Response: Thanks for your suggestion, we’ve added the full name accordingly.

  • The physiological role of KIF4A should be mentioned.

Response: Thanks a lot for your advice, we’ve added the physiological role of KIF4A in the manuscript.

3) Methods and material were not properly described:

  1. a) PCR conditions and primers are missing.

Response: Thank you and we’ve added them in the manuscript accordingly.

  1. b) The RNAse R supplier is not listed.

Response: Thank you and we’ve added it in the manuscript accordingly.

  1. c) In the case of actinomycin D, cells were seeded in 6-well plates. However, there is no information whether the treatment was done immediately or whether the cells were first cultured at some time interval and then treated with actinomycin D.

Response: Thanks for your comments. Cells were seeded in 6-well plates and treatment of actinomycin D was done immediately since NKTCL cells are suspended cells. We’ve added the information in the manuscript.

  1. d) How many cells were seeded for RNA immunoprecipitation and how long were they cultured before transfection?

Response: Thanks for your comments. 3×106 NKTCL cells were seeded on a 6cm plate, and co-transfection was performed immediately (since NKTCL cells are suspended cells) for 48h. Four plates of cells in each group were harvested for following immunoprecipitation. We’ve added the information in the manuscript.

  1. e) RNA complex purification method and primers for miR-1231, PDK1 and BCL11A are missing.

Response: Thank you and we’ve added them in the manuscript accordingly.

  1. f) MS2bs-circKIFAmt should be described/explained.

Response: Thank you and we’ve added them in the manuscript accordingly.

  1. g) Antibodies used in the immunohistochemistry should be specified.

Response: Thanks for your comments. The antibodies we used were primary antibodies of PDK1 and BCL11A (1:100, Abcam, UK) and HRP-conjugated secondary antibodies (Thermo Scientific), which has been mentioned in the manuscript.

  1. h) siRNA sequences are missing.

Response: Thanks for your comments. We’ve added the siRNA sequences in the manuscript accordingly.

4) More detailed information about the patients and their therapy is missing. It is also unclear whether the samples were obtained before, during, or after therapy. If after, information about the treatment regimen should be given.

Response: Thanks for your constructive advice. The samples were obtained before therapy.

5) More information should be provided on RNAse R resistance analysis. How was the circularity of KIF4A confirmed?

Response: Thanks for your constructive advice. RNase R digests linear RNA but not circular RNA,RNase R resistance analysis and digestion analysis using actinomycin D confirmed that circKIF4A was more stable than linear KIF4A mRNA, which confirmed the circular structure of circKIF4A.

6) In the case of Fig. 1C, the units are missing.

Response: Thanks for your suggestion. We’ve added the units in Fig. 1C.

7) NKTL is not B-cell lymphoma. Why were B-symptoms used in the COX analysis?

Response: Thanks for your comments. B-symptoms are non-specific systemic symptoms (usually including unintentional weight loss, fever, and night sweats) that can be associated with the presence of an underlying lymphoma, including various kinds of non-Hodgkin’s lymphoma and Hodgkin’s lymphoma. It has been reported that B-symptoms were related to the prognosis of NKTL.[1]

8) The authors reported that COX multivariate analysis showed that circKIF4A expression, clinical stage, ECOG score, LDH levels and metastatic status were independent prognostic indicators of NKTL. However, p<0.05 was detected in case of multivariate analysis only for circKIF4. In addition, there are several values of 0.000 in Table 1.

Response: Many thanks for your constructive comments and suggestions. We’ve checked the data again and it’s a descriptive mistake in the manuscript. We’ve revised the report to only circKIF4A expression being the independent prognostic indicator. The values of <0.001 in table 1 were mistakenly recorded as 0.000in Table 1, and we’ve revised it.

9) Fig. 2: Units for OS, PFS (x-axis) and probability of survival (y-axis) are missing.

Response: Thanks for your suggestion. We’ve added the units in Fig. 2.

10) The reason why the authors studied glucose and lactate production should be explained and the methods of glucose and lactate production study should be described in the methodology section.

Response: Thanks for your suggestion. We performed the glucose uptake and lactate production to show that silencing circKIF4A inhibits glycolysis of NKTL cells, which is a major source of energy metabolism in tumor cell growth. The methods were also added in the manuscript.

11) Fig. 3: It is not explained what NC, A#1 and A#2 mean.

Response: Thanks for your comments. Si-circNC means negative control siRNA, 5’-UUCUUCGAACGUGUCACGUTT-3’(sense); 5’-ACGUGACACGUUCGGAGAATT-3’(antisense); si-circKIF4A#1 and si-circKIF4A#2 are two siRNAs we designed to inhibit the expression of circKiF4A. We’ve added the description in the Figure.3 legend.

12) The reason why the authors focused on miR-1231 should be explained.

Response: Thanks for your suggestion. We used TargetScan to predict miR-1231 binding sites, and obtained 7 candidate miRNAs with site-type 7mer-m8 and context+ score percentile greater than 90 as screening criteria. Then we detected their expression in NKTL cell lines, miR-1231 was the most significantly downregulated miRNA.

13) MS2bs-circKIF4A-mut and PDK1 and BLC11A luciferase reporter mutations should be specified.

Response: Thank you and we’ve added them in the manuscript accordingly.

14) Fig.  6. More information about cases 1-3 should be provided. Samples were taken from 227 NKTCL patients. It is not clear what cases 1-3 mean.

Response: thanks for your comments. Cases 1-3 are three representative samples for detection of EBER, Ki-67, PDK1 and BCL11A by IHC.

Reviewer 2 Report

qPCR denotes quantitative PCR, no need to mention qRT-PCR.

Fig.2 graphs are not legible. B symptoms can be briefly explained for the readers in a single sentence.

Fig 4 D table is not legible. 4E F : Graph titles and legends are outside the panels. Fig.4 panel is not aligned to main text.

Graphs in this paper needs to be scaled to fit uniformly.

Fig 6: C: Check the histology magnification denoted. The images need to be edited and quantified prior to interpretation.

Author Response

Review2

Comments and Suggestions for Authors

qPCR denotes quantitative PCR, no need to mention qRT-PCR.

Response: Thanks for your advice, we’ve revised them in the manuscript.

Fig.2 graphs are not legible. B symptoms can be briefly explained for the readers in a single sentence.

Response: Thanks for your comments. The graphs were compressed in the Word documents, and we’d provide legible original graphs to the journal for publication. The explanation of B symptoms was added in the manuscript.

Fig 4 D table is not legible. 4E F : Graph titles and legends are outside the panels. Fig.4 panel is not aligned to main text.

Graphs in this paper needs to be scaled to fit uniformly.

Response: Thanks for your comments. The graphs were compressed for Fig 4 D table in the Word documents, and we’d provide legible original graphs to the journal for publication. We’ve rearranged the graphs, panels and the main text to fit uniformly.

Fig 6: C: Check the histology magnification denoted. The images need to be edited and quantified prior to interpretation.

Response: Thanks for your comments. We’ve quantified the images in supplementary Figure 1 by image J analysis.

Reviewer 3 Report

In this work, the author demonstrated that in NKTL circ-KIF4A is highly expressed and correlates with poor NKTL prognosis, while miR-1231 is down-expressed probably recruited by circ-KIF4A functioning as a miR-sponge. The authors proposed that circ-KIF4A may influence NKTL through modulating PDK1 and BCL11A expression by functioning as a ceRNA for miR-1231.

The manuscript is presented in a well-structured manner, the experimental design is appropriate and complete to test the hypothesis, the case number used for OS and PFS is good for statistical analysis, the field is new and interesting. The weakness of the work is that the choice of circ-KIF4A and miR1231 is not due to a whole screening pointing at these two factors as relevant, but to their role in other neoplasia.

Comments:

Q1: In the Discussion, I suggest to report if there are screening of the entire circRNA and miRNA patterns in NKTL on literature, that would strengthen the importance of circKIF4A and mir1231 in this disease. I suggest also to discuss if there are other circRNAs/miRNAs important in NKTL. 

Q2: Seen that circKIF4A is a sponge also for mir127 in ovarian cancer, is this mirna taken into consideration? Have other mirnas been evaluated but not included in the paper? if it is so, why?

Q3: Fig. 1: Why in the y axis of panel A is the scale based on 1.1 and not simply 1? In panel C please specify the unit of measurement (hours) of the x axis. In the caption of panel B is missing the reference to mRNA of KIF4A. Panel C, how many experiments have been done in each cell line? The number of experiments must be included in the caption or in methods section (not only for figure 1 but also for the others results) 

Q4: Since it is used as y axis variable, can you explain how the probability of survival changes for NKTL patients, ? Which is the time range you considered for OS and PFS?

Q5: Please, in figure 3 add in the caption the meaning of si-circNC, and specify that you use two different siRNAs (#1 and #2) (correct?). Similarly, check in the other figure captions if there are abbreviations to express in full. 

Q6: Figure 3, panel B, uniform the dimensions of the legend as in the other panels.

Q7: Please, specify how you measure glucose uptake and lactate production and why you measure these parameters.

Q8: Figure 5, panels A-B: the bars are not aligned with sequences.

Q9: Figure 6, panel C: please explain in caption why you use Ki67 and EBER as markers.

Q10: The discussion of the results collected in this work could be improved if amplified, for example adding the explanation of the interaction between miR-1231 and PDK1 and BCL11A and the importance of your findings in the context of NKTL.

Author Response

Review3

Comments:

Q1: In the Discussion, I suggest to report if there are screening of the entire circRNA and miRNA patterns in NKTL on literature, that would strengthen the importance of circKIF4A and mir1231 in this disease. I suggest also to discuss if there are other circRNAs/miRNAs important in NKTL. 

Response: Thanks for your comments. Circular RNAs were rarely reported in NKTL, and no screening of the entire circRNA patterns was found in NKTL on literature. This is the first time to discuss the importance of circKIF4A and miR-1231 in this disease. We used TargetScan to predict miR-1231 binding sites, and obtained 7 candidate miRNAs with site-type 7mer-m8 and context+ score percentile greater than 90 as screening criteria. Then we detected their expression in NKTL cell lines, miR-1231 was the most significantly downregulated miRNA.

Q2: Seen that circKIF4A is a sponge also for mir127 in ovarian cancer, is this mirna taken into consideration? Have other mirnas been evaluated but not included in the paper? if it is so, why?

Response: Thanks for your suggestion. We used TargetScan to predict miR-1231 binding sites, and obtained 7 candidate miRNAs with site-type 7mer-m8 and context+ score percentile greater than 90 as screening criteria. Then we detected their expression in NKTL cell lines, miR-1231 was the most significantly downregulated miRNA. Since the miR127 was evaluate as less favorable one, we did not take it into consideration.

Q3: Fig. 1: Why in the y axis of panel A is the scale based on 1.1 and not simply 1? In panel C please specify the unit of measurement (hours) of the x axis. In the caption of panel B is missing the reference to mRNA of KIF4A. Panel C, how many experiments have been done in each cell line? The number of experiments must be included in the caption or in methods section (not only for figure 1 but also for the others results) 

Response: Thanks for your constructive suggestions. We’ve revised the Fig. 1 panel A-C accordingly. Each experiment was repeated for at least three times for the results, and we’ve added the statement in the methods section.

Q4: Since it is used as y axis variable, can you explain how the probability of survival changes for NKTL patients, ? Which is the time range you considered for OS and PFS?

Response: Thanks for your comments. The probability of survival decreased with the extension of follow-up time. Group with poor clinical features has lower survival probability. OS (overall survival) was defined as period from diagnosis of NKTL to death or loss to follow-up. PFS was defined as period from diagnosis to disease progress or death.

Q5: Please, in figure 3 add in the caption the meaning of si-circNC, and specify that you use two different siRNAs (#1 and #2) (correct?). Similarly, check in the other figure captions if there are abbreviations to express in full. 

Response: Thanks for your suggestion. si-circNC group was transfected with negative control siRNA, 5’-UUCUUCGAACGUGUCACGUTT-3’(sense); 5’-ACGUGACACGUUCGGAGAATT-3’(antisense). We designed two siRNA to inhibit the expression of circKiF4A. We’ve added them in the caption of Figure. 3.

Q6: Figure 3, panel B, uniform the dimensions of the legend as in the other panels.

Response: Thanks for your suggestion. We’ve revised it accordingly.

Q7: Please, specify how you measure glucose uptake and lactate production and why you measure these parameters.

Response: Thanks for your suggestion. We measured glucose uptake and lactate production according to manufacturer’s protocols with test kits from Promega. We measured these parameters to illustrate the inhibition of glycolysis by silencing circKIF4A in NKTL cells. Glycolysis is a major source of energy metabolism in tumor cell growth. We’ve added the above illustrations in the manuscript.

Q8: Figure 5, panels A-B: the bars are not aligned with sequences.

 Response: Thanks for your comments. Figure 5, panels A-B are from the TargetScan database.

Q9: Figure 6, panel C: please explain in caption why you use Ki67 and EBER as markers.

Response: Thanks for your suggestion. Ki67 is a nuclear protein that is associated with cellular proliferation. NK/T cell lymphoma is an aggressive lymphoma associated with Epstein-Barr virus (EBV) infection. Therefore, we used Ki67 and EBER as bio-marker of cancer cells.

Q10: The discussion of the results collected in this work could be improved if amplified, for example adding the explanation of the interaction between miR-1231 and PDK1 and BCL11A and the importance of your findings in the context of NKTL.

Response: Thanks for your valuable suggestion. We’ve improved the content in the discussion section. Our data show that circ-KIF4A is highly expressed in NKTL and that it is a predictor of poor NKTL prognosis. We found PDK1 and BCL11A as targets of miR-1231 in NKTL, and the transfection of miR-1231 inhibits the expression of PDK1 and BCL11A. Circ-KIF4A potentially regulates the expression of PDK1 and BCL11A by sponging miR-1231. All in all, our findings indicate that circ-KIF4A may influence NKTL tumor growth through modulating PDK1 and BCL11A expression by functioning as a ceRNA for miR-1231. Therefore, circ-KIF4A may become a potential prognosis predictor and therapeutic targets for NKTL.

Round 2

Reviewer 1 Report

1) Antibodies should be specified by indicating the exact type of antibody. In the case of Abcam, this is usually abXXXXX (X stands for number). The provider may have multiple different antibodies for the same target.

2) It is not clear why just cases 1 - 3 were chosen for Fig. 6. Are these cases different from the rest of the samples? Did the authors analyze any of the other samples?

Author Response

Reviewer 1

Comments and Suggestions for Authors

  • Antibodies should be specified by indicating the exact type of antibody. In the case of Abcam, this is usually abXXXXX (X stands for number). The provider may have multiple different antibodies for the same target.

Response: Thanks for your comments. The exact types of antibodies in our study were as shown below: Anti-PDK1 (ab202468), Anti-BCL11A(ab191401),Anti-Ki67(ab16667).These contents have been supplemented in the article.

2) It is not clear why just cases 1 - 3 were chosen for Fig. 6. Are these cases different from the rest of the samples? Did the authors analyze any of the other samples?

Response: Thanks for your comments. Immunohistochemical analysis of EBER, Ki-67, PDK1 and BCL11A was also done in other cases. Cases 1-3 are three representative samples for detection of EBER, Ki-67, PDK1 and BCL11A by IHC.

Reviewer 2 Report

Supplementary Fig1: Graph C and D - The significance is compared to the group needs to be shown with arrow and put the exact P values instead of asterisk. 

Fig.3 A B C D and Fig. 4 C E F G: Graphs should also denote the significance vs which group and P values. Legend should mention the statistical test performed.

Author Response

Reviewer 2

Comments and Suggestions for Authors

Supplementary Fig1: Graph C and D - The significance is compared to the group needs to be shown with arrow and put the exact P values instead of asterisk.

Response:Thanks. We have made changes according to your suggestions.

Fig.3 A B C D and Fig. 4 C E F G: Graphs should also denote the significance vs which group and P values. Legend should mention the statistical test performed.

Response: Thanks for your advice. We have made changes according to your suggestions.
